# The One Thing You Need to Change Is Emotions: The Effect of Multi-Sensory Marketing on Consumer Behavior

Moein Abdolmohamad Sagha [1], Nader Seyyedamiri [1,*], Pantea Foroudi [2] and Morteza Akbari [1]

1 Faculty of Entrepreneurship, University of Tehran, Tehran 1439813141, Iran; moinsaqa@ut.ac.ir (M.A.S.); mortezaakbari@ut.ac.ir (M.A.)
2 Department of Marketing Branding & Tourism, Middlesex University, London NW4 4BT, UK; p.foroudi@mdx.ac.uk
* Correspondence: nadersa@ut.ac.ir

**Abstract:** Retailers are increasingly aware of the importance of store atmosphere on consumers' emotions. The results of four experimental studies demonstrate that the sensory cues by which customers sense products and the amount of (in)congruency among the sensory stimuli of the products affect consumers' emotions, willingness to purchase, and experience. In the presence of moderators such as colors, jingles, prices, and scent imagery, when facing sensory-rich experiential products (e.g., juice, coffee, hamburger, soda) with different sensory cues, consumers' emotions, willingness to purchase, and experience depend on affective primacy and sensory congruency. The results (1) facilitate an improved consideration of the role of the interaction of sensory cues on customer emotions, (2) have consequences for outcomes linked with sensory congruency and affective primacy, and (3) help clarify possible incoherence in preceding studies on cross-modal outcomes in the setting of multi-sensory marketing.

**Keywords:** sensory marketing; imagery; color; price; bundling; willingness to purchase





## 1. Introduction

Marketers and managers are increasingly focusing on sensorial cues to determine consumer's definitions of emotion and experience [1–3]. Sensorial cues, such as music, color, and scent, can influence consumers' behavior [4,5]. As a result, managers should use these sensory and experimental cues to affect customers' cognitive experiences [2,6,7]. For example, although Reese sells both chocolate and peanut butter cups, the company promoted them as 'two great tastes that taste great together' to create an exclusive customer experience [8,9]. Similarly, retailers often use background music as a marketing tool [10,11]. Retailers and organizations have been progressively keen to build their marketing propositions based upon senses that determine the consumer experience, to attract customers' attention and make the experience memorable [12–14]. The increased usage of sensory cues is mainly undertaken by organizations and retailers to provide product scent, background music, user interface designs, and so on [11,15–18].

Despite the reality of such extensive sensory cues for merchandises and services, it is not certain whether and in what way different sensory systems influence others [19]. This is especially important because recently, psychology and neuroscience laboratories have found that our different senses influence our perception [20]. Moreover, although sensory marketing has seen a surge of interest in marketing and consumer psychology [1,2], less attention has been paid to the effect of multi-sensory cues on consumer behavior and experience. Delivering product sensory aspects like sound, vision, smell, and taste to consumers individually or integratively shapes the holistic consumer experience [2,13,21]. Thus, examining how the interaction between sensory cues (vision and olfaction) influences consumer emotions (arousal and pleasure) and finally affects their affective experience and willingness to purchase has both academic and real-world implications. In addition,

the results of our examination contribute to the research on the effect of multi-sensory interactions on customer behavior and experience.

As an explanatory instance, imagine that a customer sees two of the same product but with different sensory cues. Would the consumer's emotions and affective experience be more influenced by the product with congruent or incongruent background color? Would the consumer's emotions and affective experience be influenced by the product with dynamic or static olfactory/scent imagery? Would the consumer's emotions and affective experience be influenced by the product that has or that lacks background music? Would the consumer's emotions and affective experience be influenced by the product with the bundled or unbundled price?

While we are aware of only a few studies on how multi-sensory interactions influence consumer emotions and experience, there is extensive research on sensory marketing in generic terms or related to other variables (e.g., [11,17,19]). Prior studies [2,6,11,22] in sensory marketing have often studied the influence of the five senses individually on consumers' judgment and behavior. In contrast, limited research has examined the interaction between the different senses and their holistic effect on consumer emotions and experience, except for studies by Lowe and Haws [16] and Spence [23], who reported different results relating to multi-sensory marketing. For example, Lowe and Haws [16] examined the effects of acoustic pitch on consumers' perception of product characteristics over cross-modal interpretation. Spence [23] likewise described a variety of robust cross modal communications among both music and shapes and the sensory attributes of various foods and beverages. Our research findings should add to the literature of these previous investigations, as discussed in more detail in the following section.

In brief, we explore the effect of vision, olfaction, audition, and price on consumer emotions, experience, and willingness to purchase in the context of online retailing using a range of foods rich in sensory stimuli, including orange juice, coffee, hamburgers, soda, and French fries. We tested our assumptions in four varied online examinations that operated virtually with subjects from a domestic online board and college students. Study 1 examines how presenting a glass of orange juice with either a congruent or incongruent background color influences consumer emotions and experience and finally affects their willingness to purchase. Study 2 examines how presenting a coffee cup with either dynamic or static olfactory imagery influences consumer emotions and experience and finally affects their willingness to purchase. Study 3 examines how presenting a soda can either with or without background music (brand jingle) influences consumer emotions and experience and finally affects their willingness to purchase. Study 4 examines how presenting a hamburger with either a bundled or unbundled price influences consumer emotions and experience and finally affects their willingness to purchase.

Next, we review the theoretical background, and then we introduce our overall conceptualization of multi-sensory marketing as well as our hypotheses. The rest of the paper identifies and discusses the previous studies in this field. The discussion also highlights the implications for different types and roles of multi-sensory marketing as suggested by our conceptualization. We conclude with implications and possible extensions of our review and with the managerial implications, the limitations of the research, and future research directions.

## 2. Theoretical Background

Previous studies on sensory marketing have explored the setting of individual senses and consumers' perceptions and behaviors, such as the consequences of auditory degree on product cognizances, product scent and customer's appeal, and taste perception, and have focused on behaviors rather than on emotions/experience/willingness to purchase. These studies have usually introduced sensorial effects in the sense that every sensory cue shapes different perceptions and affects customer behavior in different ways [16,17,24].

The process of senses includes sensation and perception, but the sensation occurs when the sensory cue enters the receptor cell of a sense organ [1]. We suggest that when

encountering subliminal sensory cues while browsing a product in an online retail store, two opposing forces influence the consumer's willingness to purchase. First, from the affective primacy theory, an unconscious cue (i.e., an odor) induces an emotive reaction, which then progresses to the processing of a condition and adjusts its emotional assessment. Customers can consequently shape positive emotional responses to a condition with a similar previous cognitive clarification of the causes creating a positive impact [22]. Where there are multiple sensory cues, they integrate with and influence each other and then influence the process. This phenomenon occurs primarily because neuroscientists have found that some brain areas are multisensory [19]; therefore, our emotions about a product consist of the interaction between multi-senses [1]. Hence, this set of studies has proposed that multisensory interactions can affect customer decisions [25]. For instance, Ref. [26] discovered that when customers see the picture of advertised food after smelling its odor, their desire to eat increases significantly compared to those who have not seen it.

In contrast, studies in the field of sensory marketing have proposed that when the considered cue is congruent with expectations, then its attached positive effect is transferred to the overall evaluation [2,25]. Sensory marketing has been described from both the perspective of both evolutionary psychology and social neuroscience. Specifically, as a sensory stimulus evokes physiological and neurological reactions, this whole procedure influences how the sensation is described theoretically [27,28]. For example, when we are exposed to a food odor, first the senses relate to that cue and send information to the brain region that processes such sensory cues. After identification of the cue, the sensorial cues are delivered to the prefrontal cortex, where the value of their reward is evaluated (e.g., arousal or pleasure) [6]. However, given that sensory interactions and sensory congruence would create similar forecasts in various circumstances (as well as in the settings of our research) and are occasionally used conversely in the articles [1], although, there are a variety of concepts in sensory marketing. Sensory interactions refer to the stimuli presented in one sensory modality influencing another sensory modality for processing stimuli [19]. In contrast, sensory congruence is when two sensory cues are congruent, and they improve probable behaviors or reactions; for example, congruency between scent and music in a retail store may improve store evaluations [1]. In other words, sensory interactions always happen, whereas sensory congruence may happen or may not. For example, when you go shopping, you see products in a setting with sounds and scents, but those sounds and scents may or may not have congruence with the products you see.

In short, the conceptual model connected to cross-modal effects would foresee sensory interaction outcomes (i.e., stimuli in one sensory modality can compensate for or improve the desire associated with the other sensory modality), whereas affective primacy theory would only speculate about sensory congruence outcomes (i.e., priority for a product with incongruent or congruent sensory cues). Which of these hypothetical outcomes will be dominant when consumers are exposed to sensory-rich empirical products? We suggest that it will be the response related to the level of (un)likeness among the sensory stimuli faced in the online retail atmosphere. When the encountered sensory stimuli are alike, sensory marketing is likely to be a superior influence because they reinforce each other. As a result, when viewing a product in the online retail atmosphere, if another sensory cue is encountered (such as sound or imagined scent), it can improve your emotions, experience, and willingness to make a purchase. In contrast, when the encountered sensory cue is incongruent, it can decrease your emotions, experience, and willingness to make a purchase. Consequently, there will be an additional impact on sensory marketing. Therefore, when exposed to products in an online retail atmosphere, multiple sensory cues may affect us.

This research sets out to examine how multi-sensory cues influence consumers' product responses (i.e., affective responses, emotions, and willingness to purchase). The authors identify two theories (affective primacy and sensory congruence) which might explain how multi-sensory cues will influence product responses. We aim to test our hypothesis across four separate studies. In study 1, we explore the interaction of color and vision within the domain of product package (see Table S1 in Supplementary Materials). In study 2, we

demonstrate the effects of imagining a coffee scent while looking at an online retail store. Study 3 focuses on sound (how brand jingle would increase emotions and experiences and willingness to purchase). In study 4, we look at the effect of bundling on costumer emotions—with versus without access to a bundled picture of the food (see in Appendix A).

## 3. Study 1: Imagining the Color

In study 1, we tracked customer visual attraction with a moderating color with two same products (orange juice) but with either congruent or incongruent background color. We chose an orange juice brand based on an article of Hoegg and Alba [24]. We chose yellow and blue colors based on the opponent theory of color processing. Opponent processes enhance color contrasts at the boundaries of objects, such that yellow on one side of a boundary intensifies responses to blue on the other side of a boundary. On the other hand, we chose yellow because from the visual coherence perspective two color relationships should increase esthetic preference. First, two colors could be identical matches (i.e., the same point in the color space). Exact matches among component parts of a product help unify the design. Second, two colors could be distinct but closely related [29,30]. In terms of the level of arousal, the yellow hue is exciting and the blue hue is relaxing. Yellow conveys arousal, cheerfulness, confidence, creativity, excitement, extraversion, friendliness, happiness, optimism, self-esteem, sincerity, smile and spirit. Blue conveys calmness, comfort, competence, coolness, dignified duty, efficiency, intelligence, logic, peace, reflection, relaxation, reliability, security, serenity, sooth, success, tenderness, tranquility and trust. We also chose blue because it is definitely distinct from orange juice color. Identical colors have zero distance, closely related colors have small distances, distinct colors are moderately distant, and contrastive colors have large distances. The optimal arousal perspective, however, predicts an inverted-U shape, which is consistent with a preference for distinct colors [29]. We predicted that a visual sensory cue enhances the emotions, experience and willingness to purchase the product only if the color of the background is congruent with the product, because congruent color is typically the same as product color(i.e., blue is incongruent to orange juice, so:

**Hypothesis 1 (H1).** *When viewing a product with a background color, customers' emotions (H1a), affective experience (H1b), and willingness to purchase (H1c) will be greater with a congruent background color.*

### 3.1. Participants and Design

Four hundred students (167 males, 233 females; 43% aged 26–40 years) participated in the study. They were randomly assigned to conditions of a between-subjects design with two identical goods (orange juice) but with both congruent and incongruent sensory stimuli (background color: blue vs. yellow). Participants completed the study on their computers or smart phones. First, they watched a product with a background that was either congruent or incongruent. The website randomized the product in which the participants watch to each product. Afterward, participants responded to a series of questions, including the key dependent variable of willingness to purchase ("I consider the product as my first choice compared to other products").

### 3.2. Procedure

We performed a pilot study (n = 60) to confirm that the questionnaire was reliable and valid (see in Table 1). Participants saw the product picture and indicated their vision conception on a 7-point scale, as in previous studies (e.g., Ref. [11], 1 = "strongly disagree" and 7 = "strongly agree"). The outcomes of the pilot study test showed that the questionnaire was reliable and valid. To estimate validity, Cronbach's alpha of all constructs was extracted, and questions with a low alpha were deleted. We also removed some questions due to participant perception regarding the translation to Persian. In the main experiment, participants were invited to complete an online survey. First, they answered four demo-

graphic questions, and after that, they were asked to choose one product (A or B). Second, they viewed the product picture and answered 23 questions based on their product choice (see Table S1).

**Table 1.** Reliability Report.

| Items | Vision | Color | Willingness to Purchase | Arousal | Pleasure | Experience |
|---|---|---|---|---|---|---|
| $\alpha$ of Blue | 0.9 | 0.7 | 0.9 | 0.9 | 0.9 | 0.9 |
| $\alpha$ of Yellow | 0.9 | 0.8 | 0.8 | 0.8 | 0.9 | 0.8 |

*3.3. Results*

We conducted a t test to examine whether the background color affected participants' emotions, experience and willingness to purchase. The analysis revealed that costumers' emotion, experience and willingness to purchase when the background color was incongruent was significantly greater than when the background color was congruent with product (see in Table 2).

**Table 2.** Hypothesis testing results.

| | | Overall | | | Male | | | Female | | |
|---|---|---|---|---|---|---|---|---|---|---|
| | Group | Mean | t | Sig. (2-tailed) | Mean | t | Sig. (2-tailed) | Mean | t | Sig. (2-tailed) |
| Vision | Blue | 2.74 | 8.529 | 0.000 | 2.67 | 4.619 | 0.000 | 2.79 | 7.226 | 0.000 |
| | Yellow | 1.93 | | | 1.99 | | | 1.87 | | |
| Color | Blue | 3.04 | 11.520 | 0.000 | 2.92 | 6.749 | 0.000 | 3.11 | 9.312 | 0.000 |
| | Yellow | 2.00 | | | 1.96 | | | 2.02 | | |
| Willingness to Purchase | Blue | 2.58 | 7.486 | 0.000 | 2.46 | 4.084 | 0.000 | 2.66 | 6.278 | 0.000 |
| | Yellow | 1.88 | | | 1.87 | | | 1.88 | | |
| Arousal | Blue | 2.60 | 7.748 | 0.000 | 2.55 | 4.127 | 0.000 | 2.63 | 6.674 | 0.000 |
| | Yellow | 1.86 | | | 1.93 | | | 1.81 | | |
| Pleasure | Blue | 2.71 | 8.215 | 0.000 | 2.63 | 4.217 | 0.000 | 2.77 | 7.200 | 0.000 |
| | Yellow | 1.94 | | | 2.00 | | | 1.89 | | |
| Experience | Blue | 1.88 | 6.909 | 0.000 | 1.81 | 3.283 | 0.001 | 1.93 | 6.301 | 0.000 |
| | Yellow | 1.38 | | | 1.43 | | | 1.33 | | |

The results demonstrate the value for emotions was significantly greater for the incongruent background color condition ($M_{Arousal}$ = 2.60, $SD$ = 0.996, $M_{Pleasure}$ = 2.71, $SD$ = 0.935) than for the congruent background color condition ($M_{Arousal}$ = 1.86, $SD$ = 0.892, $M_{Pleasure}$ = 1.94, $SD$ = 0.949). The data support H1a. The experience was greater for the incongruent background color condition ($M_{Experience}$ = 1.88, $SD$ = 0.766) than for the congruent background color condition ($M_{Experience}$ = 1.38, $SD$ = 0.695), supporting H1b. The background was discovered to have meaningfully contrasting relationships for H1c. As expected in H1c, background color had a stronger impact on willingness to purchase for the incongruent background color condition ($M_{WTP}$ = 2.58, $SD$ = 0.985) than for the congruent background color condition ($M_{WTP}$ = 1.88, $SD$ = 0.892). Furthermore, all these effects were greater for the women than the men.

*3.4. Discussion*

The outcomes of Study l demonstrate that when viewing a product with a background color, customers' emotions, affective experience, and willingness to purchase will be greater when the background color is incongruent than when the background color is congruent.

## 4. Study 2: Imagining the Smell

In study 2, we investigate mental imagery of scents on consumer emotional responses. We choose coffee smell based on [31]. Coffee smell is well known and based on this we choose it. Further, we choose a purple cup because it conveys authenticity, charming, dignified, exclusive, luxury, quality, regal, sensuality, sophistication, spiritual, stately and upper class. From a theoretical point of view, if palatable food odors can elicit consumer approach behaviors such as salivation and desire to eat, would imagined odors do the same? Imagined odors will elicit consumer approach responses when the consumer can create a clear visual mental image of the odor referent (the object that emits the odor). One can mentally "smellize" an object without smelling the object itself (i.e., engage in olfactory imagery). Engaging in olfactory imagery facilitated the recognition of odors, just as engaging in visual imagery facilitated the recognition of pictures. The motor component of olfactory imagery such as the act of sniffing contributes to the vividness of olfactory images. The evidence to date on olfactory imagery suggests that the effects of imagined odors are similar to those of odors that are sensorially perceived [32]. We thus choose a coffee gif with vapor to stimulate smell imagery. Dynamic scent in our work is defined as an animation related to a scent that stimulates olfactory receptors. In another hand, static scent is a static image related to a scented product.

**Hypothesis 2 (H2).** *When viewing a product with an imagined scent, customers' emotions (H2a), affective experience (H2b), and willingness to purchase (H2c) will be greater with a dynamic imagined scent rather than static imagined scent.*

### 4.1. Participants and Design

Four hundred students (167 males, 233 females; 43% between 26 and 40 years) participated in the study. They were randomly assigned to conditions of a between-subjects design with two identical goods (a cup of coffee) but with different sensory stimuli (scent imagery: dynamic vs. static). Participants completed the study on their computers or smart phones. First, they watched a product with or without a vapor animation. The website randomized the product which the participants watch. Afterward, participants responded to a series of questions, including the key dependent variable of willingness to purchase ("I consider the product as my first choice compared to other products").

### 4.2. Procedure

We performed a pilot study (n = 60) to confirm that the questionnaire was reliable and valid (see in Table 3). Participants viewed the product picture and indicated their vision conception on a 7-point scale, as in previous studies (e.g., Ref. [11], 2019; 1 = "strongly disagree", and 7 = "strongly agree"). The outcomes of the pilot study showed that the questionnaire was reliable and valid. To estimate validity, Cronbach's alpha of all the constructs was extracted, and questions with a low alpha were deleted. We also removed some questions due to participant perception regarding the translation to Persian. In the main experiment, participants were invited to complete an online survey. First, they answered four demographic questions, and after that, they were asked to choose one product (A or B). Second, they viewed the product picture and answered 27 questions based on their product choice (see Table S1).

**Table 3.** Reliability Report.

| Items | Vision | Olfaction | Willingness to Purchase | Arousal | Pleasure | Experience |
|---|---|---|---|---|---|---|
| α of Coffee with Vapor | 0.8 | 0.7 | 0.7 | 0.8 | 0.8 | 0.8 |
| α of Coffee without Vapor | 0.7 | 0.7 | 0.7 | 0.7 | 0.7 | 0.7 |

*4.3. Results*

We conducted a t test to examine whether the imagined scent affected participants' emotions, experience and willingness to purchase. The analysis revealed that costumers' emotion, experience and willingness to purchase when the imagined scent was dynamic was significantly greater than when the imagined scent was static (see in Table 4).

**Table 4.** Hypothesis testing results.

| | | Overall | | | Male | | | Female | | |
|---|---|---|---|---|---|---|---|---|---|---|
| | **Group** | **Mean** | **t** | **Sig. (2-tailed)** | **Mean** | **t** | **Sig. (2-tailed)** | **Mean** | **t** | **Sig. (2-tailed)** |
| Vision | Coffee with Vapor | 4.12 | 16.061 | 0.000 | 3.96 | 8.167 | 0.000 | 4.23 | 14.398 | 0.000 |
| | Coffee without Vapor | 2.38 | | | 2.51 | | | 2.28 | | |
| Olfaction | Coffee with Vapor | 5.80 | 17.108 | 0.000 | 5.64 | 8.801 | 0.000 | 5.90 | 15.227 | 0.000 |
| | Coffee without Vapor | 3.34 | | | 3.60 | | | 3.13 | | |
| Willingness to Purchase | Coffee with Vapor | 3.10 | 14.730 | 0.000 | 2.99 | 7.585 | 0.000 | 3.17 | 13.058 | 0.000 |
| | Coffee without Vapor | 1.85 | | | 1.94 | | | 1.78 | | |
| Arousal | Coffee with Vapor | 3.18 | 14.680 | 0.000 | 3.11 | 7.585 | 0.000 | 3.22 | 13.063 | 0.000 |
| | Coffee without Vapor | 1.91 | | | 2.05 | | | 1.81 | | |
| Pleasure | Coffee with Vapor | 3.24 | 14.885 | 0.000 | 3.15 | 7.380 | 0.000 | 3.30 | 13.574 | 0.000 |
| | Coffee without Vapor | 1.95 | | | 2.13 | | | 1.82 | | |
| Experience | Coffee with Vapor | 2.39 | 14.970 | 0.000 | 2.33 | 8.060 | 0.000 | 2.42 | 12.884 | 0.000 |
| | Coffee without Vapor | 1.43 | | | 1.54 | | | 1.35 | | |

The results demonstrate that the value for the emotions was significantly greater for the dynamic scent imagery condition ($M_{\text{Arousal}}$ = 3.18, $SD$ = 0.664, $M_{\text{Pleasure}}$ = 3.24, $SD$ = 0.631) than for the static scent imagery condition ($M_{\text{Arousal}}$ = 1.91, $SD$ = 1.017, $M_{\text{Pleasure}}$ = 1.95, $SD$ = 1.045). The data support H2a. The experience was stronger for the dynamic scent imagery condition ($M_{\text{Experience}}$ = 2.39, $SD$ = 0.493) than for the static scent imagery condition ($M_{\text{Experience}}$ = 1.43, $SD$ = 0.757), supporting H2b. Support was discovered for meaningfully contrasting relationships for H2c. As expected in H2c, imagined scent had a stronger impact on willingness to purchase for the dynamic scent imagery condition ($M_{\text{WTP}}$ = 3.10, $SD$ = 0.706) than for the static scent imagery condition ($M_{\text{WTP}}$ = 1.85, $SD$ = 0.973). Furthermore, all these effects were greater for the women than the men.

*4.4. Discussion*

The outcomes of Study 2 demonstrate that when seeing a product with dynamic scent imagery, customers' emotions, affective experience, and willingness to purchase will be greater when the scent imagery is dynamic than when the scent imagery is static.

## 5. Study 3: Sound of Soda

In Study 3, we examine auditory effects with moderating brand sound with two same products (Soda soft drink) but with either presence or absence of auditory sensory cue. We choose Soda soft drink based on work of Meyerding and Mehlhose [33]. Our study is based on [34]. They say that music as a sense expression can create a sound experience and enhance a brand's identity and image. Music can also affect people's degree of arousal. Music that represents the desired level of arousal should be played to elucidate the brand's identity. For this purpose, we use Soda's brand and called Soda brand jingle. A sound brand is a sound or melody that is distinctly recognizable [34].

**Hypothesis 3 (H3).** *When viewing the product in an online retail store, customers' emotions (H3a), affective experience (H3b), and willingness to purchase (H3c) will be greater with the brand jingle.*

### 5.1. Participants and Design

Four hundred students (167 males, 233 females; 43% between 26 and 40 years) participated in the study. They were randomly assigned to conditions of a between-subjects design with two identical goods (Soda soft drink) but with either the presence or absence of sensory stimuli (background sound: brand jingle vs. no brand jingle). Participants completed the study on their computers or smart phones. First, they watched a clip—either silent or with sound—of the same product. The website randomized the product in which the participants watched. Afterwards, participants responded to a series of questions, including the key dependent variable of willingness to purchase ("I consider the product as my first choice compared to other products").

### 5.2. Procedure

We performed a pilot study (n = 60) to confirm that the questionnaire was reliable and valid (see in Table 5). Participants viewed the image of the product and indicated their vision conception on a 7-point scale, as in previous research (e.g., Ref. [11], 2019; 1 = "strongly disagree", and 7 = "strongly agree"). The outcomes of the pilot study showed that the questionnaire was reliable and valid. To estimate validity, Cronbach's alpha of all the constructs was extracted, and questions with a low alpha were deleted. We also removed some questions due to participant perception regarding the translation to Persian. The result of EFA is reported in the appendix. In the main experiment, participants were invited to complete an online survey. First, they answered four demographic questions, and after that, they were asked to choose one product (A or B). Second, they viewed the product picture and answered 23 questions based on their product choice (see Table S1).

**Table 5.** Reliability Report.

| Items | Vision | Sound | Willingness to Purchase | Arousal | Pleasure | Experience |
|---|---|---|---|---|---|---|
| α of Soda with Sound | 0.9 | 0.7 | 0.9 | 0.9 | 0.9 | 0.9 |
| α of Soda without Sound | 0.8 | 0.9 | 0.8 | 0.8 | 0.8 | 0.8 |

### 5.3. Results

We conducted a t test to examine whether the brand jingle affected participants' emotions, experience and willingness to purchase. The analysis revealed that costumers' emotion, experience and willingness to purchase when the brand jingle was absent was significantly greater than when the brand jingle was present (see in Table 6).

**Table 6.** Hypothesis testing results.

| | | Overall | | | Male | | | Female | | |
|---|---|---|---|---|---|---|---|---|---|---|
| | Group | Mean | t | Sig. (2-tailed) | Mean | t | Sig. (2-tailed) | Mean | t | Sig. (2-tailed) |
| Sound | Soda with Sound | 2.65 | −3.308 | 0.001 | 2.74 | −1.189 | 0.238 | 2.59 | −3.313 | 0.001 |
| | Soda without Sound | 2.95 | | | 2.91 | | | 2.98 | | |
| Vision | Soda with Sound | 2.49 | −4.157 | 0.000 | 2.55 | −1.914 | 0.057 | 2.45 | −3.820 | 0.000 |
| | Soda without Sound | 2.89 | | | 2.85 | | | 2.93 | | |
| Willingness to Purchase | Soda with Sound | 2.58 | −3.695 | 0.000 | 2.64 | −1.575 | 0.081 | 2.54 | −3.316 | 0.001 |
| | Soda without Sound | 2.92 | | | 2.89 | | | 2.95 | | |
| Arousal | Soda with Sound | 2.44 | −4.172 | 0.000 | 2.41 | −2.612 | 0.010 | 2.46 | −3.267 | 0.001 |
| | Soda without Sound | 2.86 | | | 2.83 | | | 2.88 | | |
| Pleasure | Soda with Sound | 2.46 | −4.381 | 0.000 | 2.40 | −2.890 | 0.004 | 2.49 | −3.313 | 0.001 |
| | Soda without Sound | 2.89 | | | 2.86 | | | 2.92 | | |
| Experience | Soda with Sound | 1.86 | −3.857 | 0.000 | 1.86 | −2.514 | 0.013 | 1.86 | −2.906 | 0.004 |
| | Soda without Sound | 2.16 | | | 2.17 | | | 2.16 | | |

The results demonstrate the value for emotions was significantly greater for the absence of brand jingle condition ($M_{\text{Arousal}}$ = 2.86, $SD$ = 0.864, $M_{\text{Pleasure}}$ = 2.89, $SD$ = 0.835) than for the presence of brand jingle condition ($M_{\text{Arousal}}$ = 2.86, $SD$ = 0.864, $M_{\text{Pleasure}}$ = 2.89, $SD$ = 0.835). The data did not support H3a. The experience was greater for the absence of brand jingle condition ($M_{\text{Experience}}$ = 2.16, $SD$ = 0.668) than for the presence of brand jingle condition ($M_{\text{Experience}}$ = 1.86, $SD$ = 0.873), not supporting H3b. No support was discovered for meaningfully contrasting relationships for H3c. As expected in H3c, brand jingle had a stronger impact on willingness to purchase for the absence of brand jingle condition ($M_{\text{WTP}}$ = 2.92, $SD$ = 0.805) than for the presence of brand jingle condition ($M_{\text{WTP}}$ = 2.58, $SD$ = 1.022). Furthermore, all these effects were greater for the women than the men.

### 5.4. Discussion

The outcomes of Study 3 demonstrate that when viewing a product without a brand jingle, customers' emotions, affective experience, and willingness to purchase will be greater when the brand jingle is absent than when the brand jingle is present.

## 6. Study 4: Cost of Hamburger

In study 4, we want to track consumer visual attention by moderating bundle pricing. We choose a hamburger sandwich for this examination based on Lowe and Haws [16]. We also choose hamburger sandwich because it is common to eat the sandwich with soda and French fries. The component outcomes may be evaluated together (consolidated) or separately (partitioned) with associated differences in perceived value. Partitioned (vs. consolidated) prices, as well as different partitions of a total price, may induce different reference comparisons and affect evaluations and choice. If prices are partitioned by component, consumers can easily add them to determine the total price of the bundle and then evaluate the associated loss. Such editing leaves the mental account on the price (loss) side identical, regardless of whether the presentation is partitioned or consolidated [35].

**Hypothesis 4 (H4).** *When viewing a product in an online retail store, customers' emotions (H4a), affective experience (H4b), and willingness to purchase (H4c) will be greater with bundled products than unbundled ones.*

### 6.1. Participants and Design

Four hundred students (167 males, 233 females; 43% between 26 and 40 years) participated in the study. They were randomly assigned to conditions of a between-subjects design with two goods from the same category (hamburger with soda and fried potato vs. hamburger) but with each having different sensory stimuli (price: bundled price vs. single price). Participants completed the study on their computers or smart phones. First, they watched a bundled or single product with a price tag. The website randomized the product and the participants watched each product. Afterward, participants responded to a series of questions, including the key dependent variable of willingness to purchase ("I consider the product as my first choice compared to other products").

### 6.2. Procedure

We performed a pilot study (n = 60) to confirm that the questionnaire was reliable and valid (see in Table 7). Participants viewed the product picture and indicated their vision conception on a 7-point scale, as in previous research (e.g., [11], 2019; 1 = "strongly disagree", and 7 = "strongly agree"). The outcomes of the pilot study showed that the questionnaire was reliable and valid. For estimating validity, Cronbach's alpha of all the constructs was extracted, and questions with a low alpha were deleted. We also removed some questions due to participant perceptions regarding the translation to Persian. The result of EFA is reported in the appendix. In the main experiment, participants were invited to complete an online survey. First, they answered four demographic questions, and after

that, they were asked to choose one product (A or B). Second, they viewed the product picture and answered 23 questions based on their product choice (see Table S1).

**Table 7.** Reliability Report.

| Items | Vision | Price | Willingness to Purchase | Arousal | Pleasure | Experience |
|---|---|---|---|---|---|---|
| $\alpha$ of Hamburger Bundle | 0.9 | 0.7 | 0.9 | 0.9 | 0.9 | 0.9 |
| $\alpha$ of Hamburger | 0.9 | 0.9 | 0.9 | 0.9 | 0.9 | 0.9 |

*6.3. Results*

We conducted a t test to examine whether the price affected participants' emotions, experience and willingness to purchase. The analysis revealed that costumers' emotion, experience and willingness to purchase when the price was bundled was significantly greater than when the price was unbundled (see in Table 8).

**Table 8.** Hypothesis testing results.

| | | Overall | | | Male | | | Female | | |
|---|---|---|---|---|---|---|---|---|---|---|
| | Group | Mean | t | Sig. (2-tailed) | Mean | t | Sig. (2-tailed) | Mean | t | Sig. (2-tailed) |
| Vision | Hamburger Bundle | 2.93 | 5.481 | 0.000 | 2.86 | 3.398 | 0.001 | 2.98 | 4.231 | 0.000 |
| | Hamburger | 2.43 | | | 2.38 | | | 2.47 | | |
| Price | Hamburger Bundle | 3.24 | 7.827 | 0.000 | 3.21 | 5.564 | 0.000 | 3.26 | 5.492 | 0.000 |
| | Hamburger | 2.53 | | | 2.44 | | | 2.60 | | |
| Willingness to Purchase | Hamburger Bundle | 3.11 | 6.083 | 0.000 | 3.04 | 4.037 | 0.000 | 3.16 | 4.471 | 0.000 |
| | Hamburger | 2.53 | | | 2.44 | | | 2.60 | | |
| Arousal | Hamburger Bundle | 3.08 | 5.824 | 0.000 | 3.03 | 3.922 | 0.000 | 3.11 | 4.231 | 0.000 |
| | Hamburger | 2.52 | | | 2.44 | | | 2.59 | | |
| Pleasure | Hamburger Bundle | 3.10 | 5.916 | 0.000 | 3.04 | 3.771 | 0.000 | 3.15 | 4.484 | 0.000 |
| | Hamburger | 2.53 | | | 2.45 | | | 2.60 | | |
| Experience | Hamburger Bundle | 2.28 | 5.477 | 0.000 | 2.24 | 3.657 | 0.000 | 2.31 | 4.003 | 0.000 |
| | Hamburger | 1.89 | | | 1.83 | | | 1.94 | | |

The results demonstrate that the value for emotions was significantly greater for the bundled price condition ($M_{\text{Arousal}}$ = 3.08, $SD$ = 0.810, $M_{\text{Pleasure}}$ = 3.10, $SD$ = 0.838) than for the single price condition ($M_{\text{Arousal}}$ = 2.52, $SD$ = 1.074, $M_{\text{Pleasure}}$ = 2.53, $SD$ = 1.080). The data supported H4a. The experience was greater for the bundled price condition ($M_{\text{Experience}}$ = 2.28, $SD$ = 0.635) than for the single price condition ($M_{\text{Experience}}$ = 1.89, $SD$ = 0.797), supporting H4b. Support was discovered for meaningfully contrasting relationships for H4c. As expected in H4c, price had a slightly stronger impact on willingness to purchase for the bundled price condition ($M_{\text{WTP}}$ = 3.11, $SD$ = 0.809) than for the single price condition ($M_{\text{WTP}}$ = 2.53, $SD$ = 1.074). Furthermore, all these effects were greater for the women than the men.

*6.4. Discussion*

The outcomes of Study 4 demonstrate that when seeing a product with a bundled price, customers' emotions, affective experience, and willingness to purchase will be greater when the price is bundled than when the price is unbundled.

## 7. General Discussion

### 7.1. Summary and Conclusions

Prior researches have tried to identify marketing strategies that positively affect consumer behavior and attain excellent customer experience to achieve business sustainability [36]. We attempted to find outcomes that ensure sustainability of the food industry. The conclusions of four online examinations—directed through different situations and clusters of customers (via online panels of subjects) and using various kinds of products (orange juice, cup of coffee, hamburger, and Soda soft drink)—show that different types of sensory cues and the (in) congruency among the sensory stimuli (link with visual) of the products affect customer emotions, experience, and willingness to purchase. We posit that consumer emotions, experiences and willingness to purchase would depend on two-color relationships. Specifically, we propose that when product and background color are incongruent, emotions will be greater and, consequently, there will be a superior experience and higher willingness to pay for the product. In contrast, when the product and background color are congruent, emotions, experience, and willingness to pay for it will be lower. We also reveal that this shape of outcomes does not hold for the same kinds of sensory stimuli. The current investigation is an initial exploration of how visual cues with the moderating factor of background color can influence consumer emotions and experience and then affect their willingness to pay.

Two contrasting theoretical effects occur when customers browse a product in an online retail store. The theory of affective primacy theory (More [22]) predicts positive emotional responses to a condition without a previous cognitive explanation of the causes establishing the positive affect, whereby subliminal cues (e.g., an odor) elicit an affective reaction, which affects the processing of a state then adjusts its emotional assessment. Therefore, affective primacy predicts positive reactions (i.e., preference for picture of product with dynamic scent). On the other hand, the theory of sensory congruency [25] predicts that the overall evaluation with a considered congruent sensory cue will increase with expectations. Thus, sensory congruency theory implies that congruent sensory cues would lead to more favorable product responses than incongruent cues. We assumed that impacts connected to affective primacy will be extra dominant when viewing products with dissimilar sensory stimuli, while sensory congruency will be leading when viewing products with the same sensory stimuli. The outcomes of our tests supported the suggested effects and as well offer evidence for the hypothesized theoretical model.

We furthermore develop some clear evidence for the suggested fundamental procedure. For instance, in Study 3, listening to the Soda jingle when viewing the product diminished the customer emotions, experience, and willingness to purchase because of cross-modal correspondence [16]. As a result, the presence of music congruent with the picture led to less positive effects compared to the lack of music. This is important because, in this study multi-sensory marketing is a practically related moderator of purchasing behavior. The outcomes of Study 1 offer further evidence for our theorization by investigating two identical products in different conditions. The outcomes illustrate that the priority design is ordinal. That is, for the two products with dissimilar sensory stimuli, subjects had the highest desire for the product with an incongruent background color rather than a congruent background color. Conversely, in Study 2, for the products with the same sensory stimuli, subjects had the highest desire for the product with dynamic scent imagery rather than static scent imagery. This design is compatible with the affective primacy effects expected for similar sensory stimuli and sensory congruency effects expected for dissimilar sensory stimuli. Overall, in these four studies, we found that congruency of sensory cues could positively affect consumer emotions, experience, and willingness to purchase when customers are faced with products with the same sensory stimuli, and that affective primacy plays a key role when customers are faced with products with dissimilar sensory stimuli. These effects differed by gender group, as we found that females are more likely to experience these effects than were males.

Managers regularly use different sensorial pictures as an efficient advertising means [8]. However, despite its popularity, little attention has been paid to the influences of multi-sensory marketing in the marketing literature, with previous studies testing such issues as impacts on memory while exhibiting products [15] and how the ambient scent might enhance consumption and purchase of products [6,22]. Nevertheless, from a perceptual viewpoint, to the best of our knowledge, ours is one of the first studies to test the impacts of multi-sensory marketing with similar and dissimilar sensory stimuli on customers' emotions, experience, and willingness to purchase. Along these lines, this is one of the first studies to test how sensory congruency and affective primacy impacts affect customers' behaviors and choices with various types of sensory stimuli.

These results also have implications for the research stream on sensory marketing and e-tail atmospheres in general. The majority of previous studies on e-tailing have investigated non-sensory stimuli and features (e.g., loyalty, brand logo, UGC) and have usually addressed results connected with judgments (e.g., product comparison). This study shows that sensory stimuli, virtual buying, and selections/priorities have both perceptual and practical implications. Notably, from a perceptual point of view, the results offer insight into customer behavior procedures when looking at several products with congruent vs. incongruent sensory stimuli. Though there is existing research on how customer selections are created, no research has examined the specific collection of sensory-linked factors that are investigated in the present study. Our results also offer an insight into how customers feel about a product and are willing to make a purchase when there are opposing consequences, such as sensory congruency and affective primacy. In addition, even though an extensive stream of research has emerged on customer multisensory experiences [7], no research has investigated how the (in)congruency of sensory stimuli can affect emotions and feelings while seeing products in an e-tail store. Therefore, our results contribute to an enhanced understanding of the function of sensory stimuli interactions on customer emotions and feelings construction.

One more important element of this study is that it resolves contradictions in the study results on bundling outcomes in the setting of e-tailing for sensory-rich experimental products. The findings of study 4 demonstrate that when customers view products with bundled price, there are affective primacy outcomes, but when customers view products with a single price, there are no affective primacy outcomes. In the context of these results, it is not surprising that [35] noticed bundling impacts, mainly for electronic devices, when subjects in their test saw consumption-related accessories. However, based upon our results, we suggest that sensory marketing may have had a leading impact on [8]'s research, as the products had the same sensory stimuli, and then they detected bundling impacts. A prior study by Janiszewski and Cunha [37] also reported bundling impacts. Even though Janiszewski and Cunha offered no particular information about the sensory effect of bundling (i.e., customers' perspective about products) used in their experiment, they did refer to an "attractiveness of an offer" impact (p. 543).

Nonetheless, they had no clear conclusions regarding the rationales underlying the impacts they detected. Therefore, essentially, sensory attractiveness, which would also result in "attractiveness of an offer", is likely to have an effect in their investigation and is compatible with the impacts we detected when subjects face products with visual sensory stimuli. In contrast, in Moon and Shugan's [38] research, the popularity of the bundled products had various effects on consumers' decisions. In the setting of our study, we suggest that through attractive visual sensory stimuli, bundling impacts were further leading for Janiszewski and Cunha [37] and Moon and Shugan's [38], and hence, they only saw bundling impacts. Even though the results of prior investigations on bundling impacts including rationale selections for affordable products might appear incongruous with sensory marketing area at first sight, the opposite can be revealed in the light of our results.

Interestingly, in Study 2, we discovered that altering the dynamicity of the olfactory imagery had an impact on customers' emotions. These results are in connection with

developing research investigations on cross-modal sensory impacts, which have shown that visual imagery can affect olfactory perception [32]. The results of Study 2 in the present investigation demonstrate that such a cross-modal impact can affect customers' emotions in an e-tail context. Further research is required to investigate this cross-modal impact in greater depth. Our investigation differs from previous studies that have investigated the influence of sensory marketing on customers' emotions, experience, and willingness to purchase (e.g., [4,20]). For example, [20] found that smell did not have a significant effect on emotions, and [4] noted similar findings with various product groups (e.g., cooling pad, heating pad). In contrast, in our study, no descriptive cues were offered to the subjects (except in the study in which we tested the impacts of bundling); instead, subjects made selections based solely on the virtual experimental viewing of the products.

### 7.2. Managerial Implications

As retail atmosphere has emerged as a competitive tool [39], it is relevant to have an enhanced awareness of how customers feel when they face products with congruent vs. incongruent sensory stimuli and how these impacts affect willingness to purchase. This issue is very relevant, as managers can regularly manage the cues of the retail atmosphere in an online retail store. The more than 350 billion products that Amazon supplies annually for sale provide evidence of its importance [40]. Sensory marketing appears in various formulae and frequently includes products with powerful sensory stimuli. For instance, in 2017, the ambient scent market earned more than $200 million in trade. Indeed, this market showed a 10% annual growth rate [22]. Similarly, retail store chains, such as ICA Sverige AB, and other supermarkets, recently decided to add more sensory labels to their food sections to persuade consumers to purchase more, and they observed positive outcomes [1]. Numerous academic studies have similarly considered pairs of short-term and long-term affirmative impacts of sensory marketing in the context of retail atmosphere [39].

Moreover, retailers such as Abercrombie & Fitch provide consumers with their brand scent [22]. Communicating with customers through sensory cues (e.g., scent) not only increases the emotions but can also enhance customer experience and willingness to purchase. Indeed, marketers have lately started to highlight how a variety of sensory cues can be associated with customer emotions and have also ascribed the increased attractiveness of products in the market place to sensory cues based on controlled studies of consumer behavior [32]. Given retailers' increasing emphasis on using multiple sensory cues, such as music [41], our results have important practical applications. Notably, organizations and firms have significant flexibility in defining the sensory cues of the product picture and the display design of the products in stores. For instance, while Unilever offered a pair of Axe and Dove deodorants, the firm designed the noise constructed by the Axe spray to be different from the noise constructed by the Dove spray [27]. As the findings of our research illustrate, the background color and music of the product and its (in)congruency with the product while it is being viewed can influence customer emotions, experience, and willingness to purchase. In addition, scent imagination and bundling prices can influence emotions, experience, and willingness to purchase. These results could be useful for businesses and firms when selecting effective display designs and background colors and music and in generating the scent imagination when customers see the products.

Online shops like Visa and NEST offer a variety of sensory cues for consumers before purchase [42]. In almost all these contexts, retailers can usually manage the sensory cues consumers encounter. Because retailers often use sensory cues to drive sales for specific products to enhance the effectiveness of their promotion campaigns [4], they can tactically locate the most effective sensory cues to affect consumers' emotions.

The food industry is another product type that relies heavily on sensory marketing. Sensorial attractiveness is a key factor for businesses in the food industry [1], and non-gustatory food-related sensory cues can lead to a considerable rise in consumption [6]. Also, foods are not consumed in isolation. Consumers often prefer to buy foods to consume together (e.g., hamburger, soda, and fried potato as a bundled package) or not (e.g.,

hamburger). Food product pictures containing scent imagery can increase consumers' physiological, emotional, and consumption responses. This can be developed to other parts of the food industry, such as restaurant menus (i.e., imagining the scent of food, drink, and other sensory-rich ingredients).

In the wider context, marketers and manufacturers are placing increasing emphasis on multisensory marketing [4]. As noted before, marketers have also placed emphasis on sensory stimuli, like the scents, visual images, and sounds related to products [6]. From previous studies, it is unclear how diverse mixtures of sensory stimuli could affect customer emotions in the retail environment, particularly when customers encounter these products. The current investigation offers improved understanding of how sensory stimuli related to atmosphere can affect customer emotions.

### 7.3. Limitations and Future Research Directions

We suggested that pairs of sensory congruency and affective primacy impacts can affect emotions when encountering products in an online retail atmosphere. The findings of our research offer support for our hypotheses. However, we did not conduct neuroscientific examinations for our studies because of the spread of COVID-19. In addition, the impacts of brand loyalty and symbolism can provide more procedure descriptions (see, e.g., [28]). More investigations are required to test the fundamental procedure in more depth and to eliminate possible alternative explanations for the results.

During our research, the products that customers viewed were from the same category. It might be worthwhile to examine the concept of multi-sensory marketing by different product categories. The present research could also be extended by examining the roles of relevant moderators, such as other senses, with other products and services or the personality traits of the consumer. Finally, we focused on emotions, experience, and willingness to purchase. Future studies might investigate the impacts of sensory marketing with diverse sensory stimuli on other dependent variables, such as consumer attitude, brand personality, and hedonic consumption. We anticipate that our investigation will inspire study in these and related regions.

**Supplementary Materials:** The following are available online at https://www.mdpi.com/article/10.3390/su14042334/s1, Table S1: Experimental design.

**Author Contributions:** Study conceptualisation, methodology, data collection, results, and discussion were conducted by M.A.S., P.F. and N.S. The reviewing and editing were carried out by P.F., supervisory N.S. and analysis were conducted by M.A.S. and M.A. All authors have read and agreed to the published version of the manuscript.

**Funding:** This research received no external funding.

**Institutional Review Board Statement:** Research approval was obtained from the Survey and Behavioural Research Ethics Committee of the corresponding author's institution.

**Informed Consent Statement:** Every participant had read the voluntary informed consent before he/she participated in the study.

**Data Availability Statement:** Not applicable.

**Acknowledgments:** Thank you very much all participants in this study.

**Conflicts of Interest:** The authors declare no conflict of interest.

## Appendix A

**Table A1.** Questionnaire items and references.

| Construct | Sub-Construct | Item | References |
|---|---|---|---|
| Vision | Attractiveness | The product picture is attractive. | [5,37] |
| | Likability | The product picture is likeable. | [25,26] |
| | Goodness | The product picture sounds really good. | [25,26] |
| | Desirability | In my opinion, the product picture is desirable. | [16] |
| Color | Mood | The color of the background affects my mood. | [43] |
| | Meaningful | The color of the background is meaningful. (R) | [43] |
| | Judgement and behavior | The color of the background affects my judgment and behavior. | [43] |
| | Recognizable | The color of the background is recognizable. | [43] |
| Price | Affordability | The advertised product seems affordable. | [44] |
| | Filling | The advertised product seems satisfying. | [16] |
| | Size | The sale price compared with the product size is correct. | [16] |
| | Desirable | The product in terms of price is desirable. | [35,45] |
| Music | Enjoyable | The music I heard is enjoyable. | [5] |
| | Familiar | The music I heard is familiar. | [16] |
| | Consonant | The music I heard is related to the product. | [16,41] |
| | Stimulating | The music I heard is stimulating. (R) | [46] |
| Scent Imagery | Exciting | The imagery which occurred while I watched the product picture is exciting. | [5,47] |
| | Clearness | The scent imagery which occurred while I watched the product picture is clear. | [48] |
| | Detailed | The scent imagery which occurred while I watched the product picture is detailed. | [48] |
| | Vividness | The scent imagery which occurred while I watched the product picture is vivid. | [48] |
| | Well-defined | The scent imagery which occurred while I watched the product picture is well-defined. | [48] |
| Olfaction | Scent likability | The product is likeable if it contained scent. | [25,27] |
| | Goodness | The scent I imagined of the product is good. (R) | [22,27] |
| | Attractive | The scent I imagined of the product is attractive. (R) | [22] |
| | Satisfying | The scent I imagined of the product is satisfying. | [25] |
| | Wide awake | The scent I imagined of the product is wide awake. | [25] |
| | Familiarity | The scent I imagined is familiar to me. | [22] |
| | Smell temperature | The scent I imagined is a very warm smell. | [15] |
| | Imaginable | The scent of the product is imaginable. | [26] |
| Emotion (Arousal) | Relaxing | The scent I imagined of the product is relaxing. | [15,22] |
| | Lively | The scent I imagined is lively. | [22] |
| | Bright | The scent I imagined is bright. | [22] |
| | Arousing | The scent I imagined of the product is arousing. | [22] |

**Table A1.** *Cont.*

| Construct | Sub-Construct | Item | References |
|---|---|---|---|
| Emotion (Pleasure) | Happy | The scent I imagined of the product is happy. | [5,22,47] |
| | Contenting | The scent I imagined of the product is contenting. | [25] |
| | Pleasing | The scent I imagined of the product is pleasing. | [25] |
| | Pleasantness | The product picture is pleasant. | [5,49] |
| Customer Affective Experience | Feelings and sentiments | The product induces feelings and sentiments. | [10,50] |
| | Entertainment | Viewing the product provides entertainment. | [10,50] |
| | Pleasurable | Viewing the product is pleasurable. | [10,50] |
| Willingness to Purchase | Click-through rate | The product persuade me to buy it. | [51] |
| | Choice comparison | I consider the product as my first choice compared to other products. | [43] |
| | Intention to buy in next purchase | I have a strong intention to buy the product in my next purchase. | [43] |
| | Intention to buy in distant future | I have a strong intention to buy the product in the distant future. | [43] |

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
