# Peer review of "The One Thing You Need to Change Is Emotions: The Effect of Multi-Sensory Marketing on Consumer Behavior"

_sustainability, doi:10.3390/su14042334_

Round 1

Reviewer 1 Report

This study examines the effect of muti-sensory marketing on consumer experience. The study potentially contributes to the literature of consumer experience, emotion, and consumer behaviour. The topic is relevant and articulation is in manuscript is commendable.

There are few suggestions for the author(s) which will further improve the quality of the manuscript.

  • Lines 50-58 are appearing surprisingly. These does not seems to have a proper flow.
  • “While we are not aware of any study on how multi-sensory interactions influen…” this should be rewritten.
  • Line 71-73 are contrasting due to detailed and brief words. Clarify it.
  • Authors have not mentioned / reviewed any Theory which they have utilized to explain the phenomena. It is a great idea to build your argument on Theory.
  • Authors have chosen blue Vs yellow. I am curious to know how black instead of blue might have changed the study results. Study 1 uses Orange juice and argument for choosing blue is about distance. How about black or certain other color which seems distant?
  • Further information is to be provided about research design e.g. what environment the respondents were asked to respond? Any criteria used for screening? Any control variables?
  • Line no 203/264/367, Cronbach alpha estimates validity???
  • Other sections were well written.
  • Language editing is required - Cross-check all references and get the paper proof-read.

Lastly, I wish authors all the best for their research.

Author Response

The One Thing You Need to Change is Emotions: The Effect of Multi-Sensory Marketing on Consumer Experience

RESPONSE TO REVIEWER COMMENTS AND SUGGESTIONS

Dear Editor

We thank the reviewers for their insightful and helpful comments on our paper. In the following pages, we have detailed our responses to all of the points made by the reviewers.

We are grateful for the time and effort of the reviewers. By making the changes which they have suggested, we believe that it is now a much-improved version, compared to the first submission. We hope that the reviewers will agree that the revised paper has reached the standard required for publication in the MDPI Sustainability.

Reviewer 1

Comment: Lines 50-58 are appearing surprisingly. These does not seems to have a proper flow.

Response: Thanks for your comment. we altered some of the sentences of that paragraph that help to its proper flow.

Comment: “While we are not aware of any study on how multi-sensory interactions influen…” this should be rewritten.

Response: Thanks for your comment. We rewrote this sentence.

Comment: Line 71-73 are contrasting due to detailed and brief words. Clarify it.

Response: Thanks for your comment. We clarified these sentences.

Comment: Authors have not mentioned/reviewed any Theory which they have utilized to explain the phenomena. It is a great idea to build your argument on Theory.

Response: Thanks for your comment. We mentioned and reviewed the affective primacy theory on page 4 and the theory of sensory congruency on page 17.

Comment: Authors have chosen blue Vs yellow. I am curious to know how black instead of blue might have changed the study results. Study 1 uses Orange juice and the argument for choosing blue is about distance. How about black or certain other colours which seems distant?

Response: Thanks for your comment. We use of the Cartesian system for colour selection, and based on Hurvich and Jameson's (1957) opponent theory of colour processing, our eyes have three types of cone receptors in which neural information are transformed by subsequent processing in the brain. Opponent processes enhance colour contrasts at the boundaries of objects, such that blue on one side of a boundary intensifies responses to yellow on the other side of a boundary.

Comment: Further information is to be provided about research design e.g. what environment the respondents were asked to respond to? Any criteria used for screening? Any control variables?

Response: Thanks for your comment. Complementary information about research design was added to the participants and design part of each study.

Comment: Line no 203/264/367, Cronbach alpha estimates validity???

Response: Thanks for your comment. The tales of validity estimation added to all studies.

Comment: Language editing is required - Cross-check all references and get the paper proof-read.

Response: Thanks for your comment. Language editing, crosscheck and proofreading were done.

Reviewer 2 Report

Proposed changes to the article:
- change of the title of the article - simplification and shortening of the studied issue;
- clarification of the goal and research questions
- in conclusion - please present the main conclusions proving the implementation of the purpose of the work and the implementation of research questions

Author Response

The One Thing You Need to Change is Emotions: The Effect of Multi-Sensory Marketing on Consumer Experience

RESPONSE TO REVIEWER COMMENTS AND SUGGESTIONS

Dear Editor

We thank the reviewers for their insightful and helpful comments on our paper. In the following pages, we have detailed our responses to all of the points made by the reviewers.

We are grateful for the time and effort of the reviewers. By making the changes which they have suggested, we believe that it is now a much-improved version, compared to the first submission. We hope that the reviewers agree that the revised paper has reached the standard required for publication in the MDPI Sustainability.

Reviewer 2

Comment: change of the title of the article - simplification and shortening of the studied issue

Response: Thanks for your comment. We changed some words of the title.

Comment: clarification of the goal and research questions

Response: Thanks for your comment. We clarified the goal and questions of our work.

Comment: in conclusion - please present the main findings proving the implementation of the purpose of the work and the performance of research questions

Response: Thanks for your comment. We added an overall conclusion of our four studies in the General Discussion part.